# Additive Manufacturing of Gold Nanostructures Using Nonlinear Photoreduction under Controlled Ionic Diffusion

**DOI:** 10.3390/ijms22147465

**Published:** 2021-07-12

**Authors:** Wera Di Cianni, María de la Mata, Francisco J. Delgado, Giovanni Desiderio, Sergio I. Molina, Alberto S. de León, Michele Giocondo

**Affiliations:** 1Consiglio Nazionale delle Ricerche, Istituto di Nanotecnologia S.S. Rende, Ponte P. Bucci 33C, 87036 Rende, Italy; giovanni.desiderio@cnr.it (G.D.); michele.giocondo@cnr.it (M.G.); 2Department of Physics, University of Calabria, Ponte P. Bucci 31C, 87036 Rende, Italy; 3Departamento Ciencia de los Materiales, I. M. y Q. I., IMEYMAT, Facultad de Ciencias, Universidad de Cádiz, Campus Río San Pedro, s/n, 11510 Puerto Real, Spain; maria.delamata@uca.es (M.d.l.M.); fran.delgado@uca.es (F.J.D.); sergio.molina@uca.es (S.I.M.)

**Keywords:** gold nanoparticles, femtosecond laser, nanofabrication, direct laser writing, two-photon absorption, two-photon photoreduction

## Abstract

Multiphoton photoreduction of photosensitive metallic precursors via direct laser writing (DLW) is a promising technique for the synthesis of metallic structures onto solid substrates at the sub-micron scale. DLW triggered by a two photon absorption process is done using a femtosecond NIR laser (λ = 780 nm), tetrachloroauric acid (HAuCl_4_) as a gold precursor, and isinglass as a natural hydrogel matrix. The presence of a polymeric, transparent matrix avoids unwanted diffusive processes acting as a network for the metallic nanoparticles. After the writing process, a bath in deionized water removes the gold precursor ions and eliminates the polymer matrix. Different aspects underlying the growth of the gold nanostructures (AuNSs) are here investigated to achieve full control on the size and density of the AuNSs. Writing parameters (laser power, exposure time, and scanning speed) are optimized to control the patterns and the AuNSs size. The influence of a second bath containing Au^3+^ to further control the size and density of the AuNSs is also investigated, observing that these AuNSs are composed of individual gold nanoparticles (AuNPs) that grow individually. A fine-tuning of these parameters leads to an important improvement of the created structures’ quality, with a fine control on size and density of AuNSs.

## 1. Introduction

Recent progress in the field of additive manufacturing (AM) in conjunction with the development of nanotechnology allowed the implementation of new nanomaterials in the industrial production fully aligned with the industry 4.0 paradigm. Advances in this research field have led to products with enhanced mechanical, optical, and electrical properties at the micro- and nano-scale. These properties can be the rather different compared with the solid macroscopic counterpart. The improvement fine tuning in the properties could lead to the production of new industrial targets and different types of devices and sensors, with potential applications from medicine to aeronautics [1,2,3,4].

At this scale, the size and the spatial configuration of produced single features start to influence the characteristics of the whole object, enabling an array of new, exotic, and otherwise unachievable properties [5]. Attention is focused especially on the fabrication of metal-containing nanomaterials, in particular Au-based. The industrial production is registering a significant demand for the fabrication of Au-containing nanomaterials because of their numerous applications in photonics, biomedicine, electronics, and optics [6,7,8,9,10,11].

Different techniques for the production of such nanomaterials include optical lithography [12], contact printing [13], and laser-induced forward transfer (LIFT) [14] structuring. As an alternative to these techniques, direct laser writing (DLW) emerges as an example of an ever-growing relevant AM optical lithography which allows micro- and nano-fabrication. DLW consists of a laser scanning fabrication technique, enabling maskless print structures using a photoresist, with a resolution depending solely on the output power and the beam waist of the used laser beam. When the optical absorption is governed by single photon processes, this technique allows for 3D patterning, with resolutions down to typically few microns, in the same order of that given by the mask lithography. However, the traditional techniques for the production of such nanomaterials involve multiple, time- and cost-intensive synthetic steps, require expensive equipment, and, in most of the cases, nanocomposites are inherently limited to two-dimensional (2D) processing [15,16,17,18,19].

It is in this direction that nano-printing techniques can take over. In particular, the two-photon lithography (TPL) and two-photon polymerization (TPP) have proven to overcome these limitations and to fabricate well-defined 3D micro/nano-structures with resolution down to 100 nm [20,21,22,23]. This is possible thanks to the two-photon absorption (TPA) process, a nonlinear third order optical phenomenon that can occur in a material between electronic states featuring the same inversion symmetry, providing that an intensity threshold is overcome, using narrowly focused pulsed lasers (100 fs) with a cross-section in the order of 10^−50^ cm^4^ s photon^−1^ [6,24,25,26]. The achievable high spatial resolution is due to the square dependence on the radiation intensity that further “narrows” the Gaussian laser beam intensity profile, along with the presence of the intensity threshold for the TPA triggering. The inner volume of the laser beam focus figure, where the TPA threshold is overcome, is the smallest polymerizable feature (voxel = volume-pixel), whose size depends on the laser intensity and on the numerical aperture (NA) of the used focusing lens. This defines the resolution of the photo-polymerization process, which also depends on the optical homogeneity of the stack crossed by the laser beam traveling from the front lens to the focal spot [6,27]. In the fabrication process, the laser focus is swept through a photoresist by a 3D stage, reproducing a 3D model made by a computer-aided design software (CAD), or analytical 2D/3D trajectories.

Combining of the two-photon photoreduction of photosensitive metallic precursors exploiting the TPA process with the DLW technique opens a promising set of conditions to in situ synthesize metallic structures onto solid substrates at the sub-micron scale. The use of different noble metal precursors allows their corresponding nanoparticles (e.g., silver, gold, platinum, or palladium) to be obtained through a photoreduction reaction. In particular, HAuCl_4_ is the most common gold precursor used for the synthesis of gold nanostructures (AuNSs) and it can be photo-reduced by single-photon UV irradiation and by TPA of infrared light [28].

Even though TPA was already discovered in 1930 by Marie Goeppert Mayer [29], it was not until the 1990s that it started to be fully exploited, discovering the optical properties and the plasmonic effects of AuNSs [18,30,31]. Since then, several authors explored the possibility of obtaining high complex 2D and 3D objects with sub-micron details for a wide range of materials and applications, from medical field for drug delivery [32], nano-heat sources [33,34], and microfluidic devices [35,36] to electronic components [37,38], design, or small satellites aiming to enhance the properties at the macroscale [4,25].

The proposed strategy, by combining TPA and DLW (TPA–DLW), allows the controlled growth of AuNSs and their direct, virtually 3D, patterning onto solid substrates with sub-micrometric resolution. Compared to other nano-fabrication techniques, such as focused ion-beam-induced deposition (FIBID), electron-beam-induced deposition (EBID), or molecular beam epitaxy, the TPA–DLW offers a much more rapid fabrication time and considerably reduced costs [9,39,40,41]. At the same time, it makes it possible to combine different materials and to integrate the metallic precursors into polymeric materials in addition to the usual photoresists for TPP [28].

In a previous work, our group elucidated the physics behind this TPA–DLW fabrication method in the case of a gold precursor (HAuCl_4_). We pointed out the diffusive process underlying the created AuNSs growth and the necessity to have a polymeric network (for instance, polyvinyl alcohol, PVA) in order to synthesize self-standing AuNSs, preventing their free diffusion [31,33]. Here, we present an alternative method for micro- and nano-fabrication of TPA–DLW of AuNSs, using a hydrogel matrix made of isinglass instead of PVA. The use of a hydrogel allows a better control of the sample hydration and, hence, of the precursor concentration, enhancing the control of the diffusive processes at the base of the AuNPs growth. Moreover, isinglass is a natural protein, which keeps in line with the principles of green chemistry. Along with the control of the laser parameters, the fine-tuning of the ionic concentration is also exploited to monitor the seeding density and the growth size of AuNSs clustering in the laser focus.

## 2. Results and Discussion

Gold nanostructures (AuNSs) were synthesized through the TPA–DLW process, taking advantage of the photoreduction with a fine-tuning of shape and size acting on the ionic concentration of the salt, using a solid opaque substrate (silicon), a natural polymeric matrix (isinglass), and an aqueous solution of a gold precursor, HAuCl_4_. For this purpose, the isinglass-coated silicon chip was first immersed in a 10^−2^ M HAuCl_4_ solution for 30 min (Figure 1a,b). This time was set quite longer than that expected from the Fickian diffusion to ensure the homogeneous diffusion of the Au^3+^ cations from the solution into the isinglass hydrogel matrix. Then, an NIR laser was focused at the interface with the Si substrate. Exploiting the TPA process, photoreduction of Au^3+^ into Au^0^ is activated (Figure 1c), causing the reduction of Au (III) into Au^0^ inside the voxel (the details of the mechanism can be found in Appendix A) [27,42]. These gold atoms are the seeds on which the synthesized gold nanoparticles (AuNPs) are grown through the autocatalytic reduction of the surrounding ions (Figure 1d). Isinglass is an inexpensive, natural protein that forms a hydrogel when dissolved in water at room temperature. This hydrogel is transparent to the wavelength of the NIR laser used and allows a better diffusion of the gold precursor compared to the PVA matrices commonly used. This grants a much better control on the AuNSs’ density and the AuNPs growth kinetics, leading to the creation of high-quality structures with a reduced size dispersion. A subsequent bath in deionized water removes the chloroauric ions from the hydrogel film, stopping the AuNPs’ growth. This method attempts to separately control the nucleation and the growth processes. However, it is important to point out that the lag between the laser exposure and the removal of the chloroauric ions should be at least in the range of 10^−2^ s to achieve a proper control of these two events. This condition could be fulfilled only using special microfluidic systems embedded in the sample holder. With this current setup, this time cannot be made shorter than typically 10 s and, as a consequence, there is an unavoidable initial autocatalytic growth of Au [43,44].

Different patterns, including isolated spots and linear structures, were tested by scanning electron microscopy (SEM). At first, an analysis of the dots size as a function of the laser power (LP) was performed for values ranging from 40 to 100 mW. At fixed values of the exposure time (ET = 100 ms), the mean values of the spot sizes are reported in Figure 2, where the expected increasing trend with the LP is confirmed. At higher values, the beam diameter of the laser increases, which causes larger voxel sizes, forming bigger AuNSs [45]. Furthermore, at high values of LP, thermal effects occur, which may also promote the reduction of the gold precursor [46]. The mean radius of the AuNSs spots varies from 1.22 ± 0.52 µm for LP = 40 mW to 1.97 ± 0.53 µm for LP = 100 mW.

Further insight in the tuning of the dots size was done by studying the influence of a longer ET (150 ms) in similar LP conditions. Figure 3a,b shows the printed AuNS for LP = 40 mW and 100 mW. They were compared to the ones manufactured at ET = 100 ms in Figure 2a–c, and the average values were presented in Figure 3c,d.

For LP = 40 mW, the structures made with ET = 100 ms have a smaller size than the ones fabricated with ET = 150 ms, even though some outliers can be observed. This could be due to a relatively low transition probability, as expected for a third order nonlinear optical process. Hence, in these low LP conditions, the photoreduction of Au^3+^ can be tuned with the ET, allowing control of the AuNSs’ size. However, for LP = 100 mW, the values are practically identical, confirming the low transition probability.

Fabrication of continuous AuNSs (e.g., lines) in different conditions was also carried out. A comparison of the thickness of the lines printed at LP 40, 80, and 100 mW using a constant scan speed (SS) of 200 µm/s is presented in Figure 4. Raising the LP determines an increase in the width of the printed lines, with mean values ranging from 1.2 to 1.6 µm when the laser power is increased from 40 to 100 mW, in analogy to the two-photon polymerization technique [47].

In order to assess the possibility of controlling the growth of the AuNSs, a second bath in HAuCl_4_ was carried out. In this case, the isinglass hydrogel film has already been eliminated and the AuNSs are fully accessible (Figure 5a), so the Au^3+^ ions in solution can be deposited onto the AuNSs surface and subsequently reduced into Au^0^ via autocatalytic reduction (Figure 5b), contributing to an increase in the sizes of the AuNSs in a controlled manner. A second washing step with deionized water, similar to that carried out after TPA–DLW, stops the second growth of the AuNSs (Figure 5c). The size of the AuNSs is controlled by means of the immersion time of the substrate in the HAuCl_4_ solution (Figure 5d). As an illustrative example, Figure 6 shows honeycomb-patterned AuNSs before (Figure 6a) and after (Figure 6b) immersion in a bath containing 10^−2^ M HAuCl_4_ for 30 min. Quantification of the size of these features in Figure 6c illustrates that there is a clear influence of the second bath in the growth of the AuNSs. These AuNSs seem to be composed of individual gold nanoparticles (AuNPs), which are likely to grow individually, even if part of them coalesce. However, it is quite difficult to distinguish the single AuNPs in the SEM images and determine their size (see, for instance, Figure 3a,b). This is especially challenging for structures made at high LP and ET conditions, as the thermal effects lead to an even higher photoreduction nearby, producing larger AuNSs.

Hence, to obtain more information on the morphology of the individual AuNPs of the linear AuNSs manufactured by the TPA–DLW process, transmission electron microscopy (TEM) analyses were carried out. These studies were conducted on a sample where the AuNSs were printed using LP = 100 mW and ET = 150 ms before and after growth in the second bath with HAuCl_4_ (see Figure 5). For each sample, an electron transparent lamella with the AuNSs and the silicon substrate was extracted by focused ion beam (FIB), as shown in the Figure 7. A Pt layer was deposited prior to FIB cut to protect the AuNSs. The lamella was several microns in width to ensure the acquisition of the cross-sections of various printed lines (depicted as gold nanowires, AuNWs, in Figure 7).

High-angle annular dark field (HAADF) and high-resolution transmission electron microscopy (HRTEM) images of the AuNSs after TPA–DLW are presented in Figure 8, where the individual AuNPs can be appreciated in more detail. As explained before, it should be noted that, in this case, there is a time gap of approximately 10 s between the AuNSs’ formation via TPA–DLW and the washing in deionized water that stops the autocatalytic growth of the AuNPs, even though the sample has not been immersed in the HAuCl_4_ bath yet. HAADF images provide chemical contrast (Z-contrast), and they indicate that areas with heavier elements are those containing AuNPs. Here, it can be observed that TPA–DLW is able to create larger structures by the coalescence of smaller individual AuNPs, leading to polycrystalline, highly faceted AuNSs, as previously observed in the SEM images in Figure 6b. HAADF (Figure 8b) and HRTEM (Figure 8d) images also show that some of these AuNSs are physically deposited onto the Si substrate surface, likely due to a stochastic heterogeneous nucleation event. Complementary electron dispersive X-ray (EDX) analyses presented in Figure 9 support the HAADF results, indicating that AuNSs consist of solely AuNPs, without the presence of oxidized species, precursor residues or any other impurities, given the absence of O and Cl in the EDX mapping signals where the AuNPs are observed. EDX mappings also show the thin SiOx native layer at the substrate surface, about 2 nm thick. As evidenced in Figure 8 and Figure 9, these AuNPs are able to self-assemble into larger micron-sized clusters. Moreover, the AuNPs formed possess different shapes (decahedral, pentagonal bipyramid, trigonal pyramidal, spherical, and cylindrical), as discussed by Polte and Turner [48,49], exhibiting a polycrystalline morphology which may facilitate the merging into larger AuNSs by photoreduction during the TPA–DLW process.

A lamella of similar dimensions was cut by FIB after the AuNSs were immersed in a 10^−2^ M HAuCl_4_ bath for 30 min (Figure 5d). Direct comparison between TEM images in Figure 8 and Figure 10 allows the evaluation of the influence of the bath in the growth of the AuNSs and AuNPs. After the second bath, individual AuNPs are not appreciated, but they have been merged by their facets and exhibit a higher polycrystalline morphology. Quantitative analysis of these AuNPs’ sizes show that they range from 50 ± 11 nm to 123 ± 35 nm (see Figure 11). These results support our previous findings in Figure 6, where a significant increase in the AuNSs’ size was also observed.

Based on the TEM results presented, an estimation of the AuNPs’ growth kinetics was carried out. It was assumed that the HAuCl_4_ is completely dissociated and, during the exposure, all the ions present in the voxel are reduced to metallic atoms instantaneously. The voxel size was approximated to a sphere of 300–1000 nm diameter in order to cover a wide range of different laser beam sizes, in agreement with experimental parameters of TPA–DLW. In these conditions, the theoretical amount of Au was calculated to be (2.7 ± 2.5) × 10^−18^ mol (see Appendix B). This value has to be considered as the upper limit for the reduced ions, as the HAuCl_4_ dissociation constant along with the probability to induce the proper electronic transitions should be considered. Nevertheless, this would go beyond the aims of the present discussion. Afterwards, the growth occurs via autocatalytic reaction [43,44], until it is stopped by the first washing bath in ultrapure water. From the AuNPs distribution in Figure 11, the number of Au moles after 10 s growth (i.e., the assumed lag between laser exposure in TPA–DLW and removal of the Au^3+^ ions) and after 30 min of a second immersion in a bath containing 10^−2^ M HAuCl_4_ can be estimated. Values of (7.2 ± 3.9) × 10^−18^ mol and (1.2 ± 0.8) × 10^−16^ mol were obtained, respectively (see Appendix B). The amount of Au obtained after DLW printing is higher than the theoretical Au amount estimated from the voxel size. Formally, this difference is ascribed to the autocatalytic growth, largely dominant over the starting AuNSs coalescence. Hence, it can be concluded that in these experimental conditions, AuNPs’ sizes increase up to around 30 nm diameter during the first 10 s after the exposure. Diffusion time in the range 10^−1^–10^0^ s/μm are typical for these systems, much shorter than the unavoidable lag between the exposure and the first bath in deionized water. As a consequence, even making this lag as short as possible, possibly by the use of a microfluidic setup, considering the characteristic water diffusion time in the hydrogel matrix, it cannot be shorter than typically a few seconds. This sets a limit to the smallest AuNPs size that can be obtained with this technique. After the second bath, an increase of up to two orders of magnitude is observed.

## 3. Materials and Methods

**Materials**. Tetrachloroauric (III) acid trihydrate (HAuCl_4_∙3H_2_O) was purchased from Sigma-Aldrich. Fish tail collagen (isinglass) was purchased from Cameo. Reagents were used as received without further purification under ambient conditions (22 ± 1 °C and 45 ± 10% RH). All solutions were prepared using deionized water.

**DLW fabrication of gold nanostructures (AuNSs)**. A thin film of isinglass was deposited by spin coating onto a silicon substrate using the FR10KPA Spin Coating System (CaLCtec S.r.L., Torino, Italy), after a stirring process (500 rpm at 40 °C) in order to get a hydrogel film of uniform thickness of about 3 µm (Figure 1a), considerably thicker than the voxel size (300 nm < d < 1 µm). The coated substrate was subsequently immersed in a 10^−2^ M HAuCl_4_ aqueous solution for 30 min to allow Au^3+^ diffusion into the hydrogel network (Figure 1b). Then, the substrate containing HAuCl_4_ was extracted from the bath and then exposed to a laser light source. Honeycomb-patterned dots and lines were fabricated as 2D structures by TPA–DLW using the Photonic Professional GT workstation (Nanoscribe GmbH, Karlsruhe, Germany) equipped with a pulsed femtosecond NIR Ti-Sapphire laser source (λ = 780 nm, pulse duration 100–200 fs, repetition rate = 80 MHz) connected to an inverted microscope (Figure 1c). The laser beam was focused on the sample through a 63× (N.A. 1.4) and a 25× (N.A. 1.1) objective in air configuration. Micro/nano structures were formed by sweeping the laser beam in X–Y directions using a galvo-scanner with different speeds. In order to create manifold structures, the following parameters were tested: laser power (LP) was ranged in all cases from 40 to 100 mW; the exposure time (ET) was ranged from 100 to 150 ms for honeycomb-patterned, isolated structures; and a scan speed (SS) of 200 µm/s was used in the case of printed lines. After the exposure to the TPA–DLW laser, the substrate was immediately immersed in a bath of deionized water that removed the metal precursor ions from the hydrogel film (Figure 1d).

**Controlled growth of AuNSs after TPA–DLW.** After the formation of the AuNSs, they may keep growing until the hydrogel film is washed with deionized water. In order to tune the size of the AuNSs, a second immersion in a 10^−2^ M HAuCl_4_ solution was carried out. This process allowed the growth of the AuNSs onto the substrate with the reduction of Au^3+^ into Au^0^, as depicted in Figure 5. After 30 min, a bath in deionized water stopped the growth process.

**Characterization and particle analysis**. SEM measurements were performed using an FEI Nova NanoSEM 450 microscope equipped with a field-emission gun for high-resolution analyses, with a concentric annular detector, capable to render Z-contrast imaging and with a FEI Quanta FEG 400 F7 eSEM (FEI, Eindhoven, The Netherlands) electron microscopes. Electron transparent lamella (ca. 100 nm thickness) were prepared in a Thermo Scientific Scios 2 DualBeam (Thermo Fisher Scientific, Berlin, Germany) focused ion beam scanning electron microscopy (FIB-SEM). A Pt layer was first deposited to protect the sample in the area of interest during the milling process. Then, the lamella was extracted and thinned by Ga ions at 30 kV. The final thinning and cleaning were carried out at 5 kV. FIB cross-section samples were analyzed by TEM and STEM (including HRTEM, STEM, HAADF-STEM, and EDX). Measurements were carried out using a Jeol J2100 (Jeol, Tokyo, Japan) provided with a LaB6 filament operated at 200 kV, and an FEI TALOS F200X (FEI, Eindhoven, The Netherlands) equipped with a field emission gun (FEG) source at 200 kV, combining high-resolution (S)TEM imaging with EDX signal detection. AuNSs and AuNPs morphology and size analysis was done using ImageJ software v1.8.0.

## 4. Conclusions

In this paper, we present a general protocol for TPA–DLW fabrication of AuNSs composed of individual AuNPs with different sizes. Enhanced quality of the AuNSs was reached, in comparison with traditional PVA matrix, thanks to the good transparency and reversible thermal behavior of the isinglass hydrogel. The biological nature of isinglass and lower HAuCl_4_ concentration herein presented shifts this technology towards a more environmentally friendly process, compared with previous works. AuNSs were obtained by optimizing different printing parameters, such as laser power, exposure time, and scan speed. These values can be tuned to modify the morphology and size of the AuNSs, proving that this method can be exploited to get a fine and practically real-time control on the size of the spots or width of the lines of the AuNSs and the diameter of the AuNPs in dependence of the parameters used. SEM and TEM results showed that these materials are printed in the shape of hierarchical structures, showing AuNSs in the range of 0.7–2.5 µm containing polycrystalline AuNPs with well-controlled spatial distribution and defined shapes with sizes around 50 nm. Moreover, the growing of the AuNSs by immersion in a HAuCl_4_ solution the size of both the AuNSs and the AuNPs to be further tuned. We foresee that this technique will allow the expansion of the uses of AuNSs and AuNPs, as well as other nanomaterials of metallic nature, such as silver, platinum, or palladium in different applications. Thanks to the optical properties of the transparent polymeric matrix and gold precursor, the use of this procedure is potentially interesting in plasmonic applications, biosensors, electronic devices, or microfluidics.

## Figures and Tables

**Figure 1 ijms-22-07465-f001:**
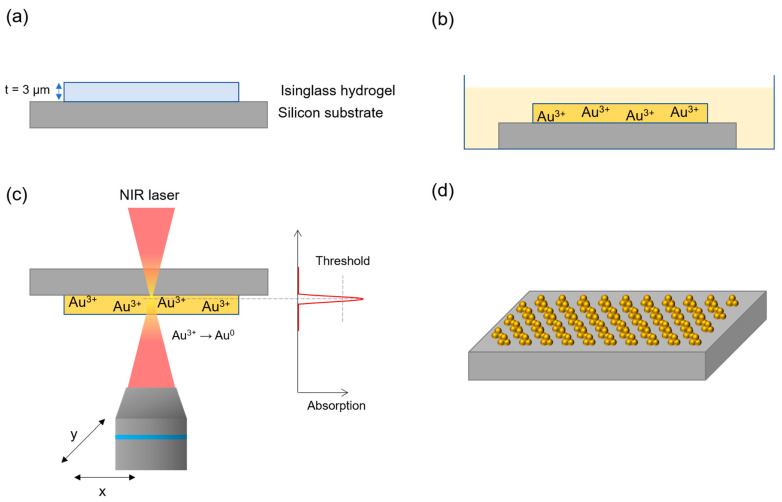
Scheme depicting the nanofabrication of AuNSs via TPA–DLW process. (**a**) a hydrogel layer is deposited on the top of a silicon substrate; (**b**) immersion of the hydrogel film in a HAuCl_4_ solution for 30 min to allow Au^3+^ diffusion into the hydrogel matrix; (**c**) photoreduction of Au^3+^ into Au^0^ via TPA–DLW using an NIR laser, taking advantage of the TPA process; (**d**) a washing step with deionized water stops the created AuNSs growth; a second washing step with warm water removes the hydrogel film, leaving the AuNSs attached to the silicon substrate.

**Figure 2 ijms-22-07465-f002:**
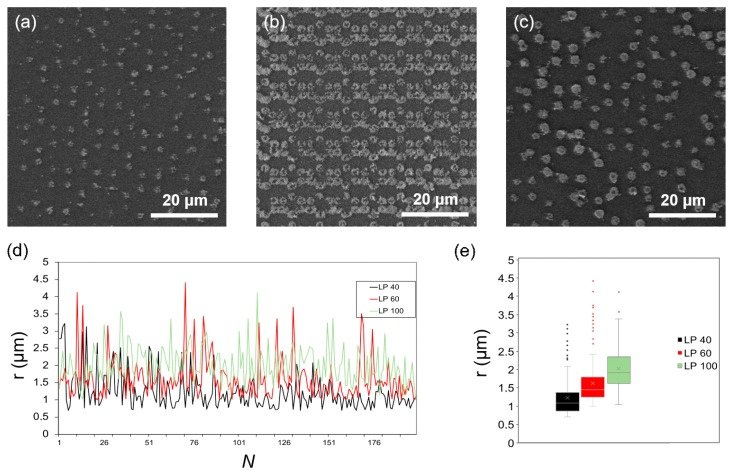
SEM micrographs showing AuNSs fabricated via TPA–DLW using an ET = 100 ms and an LP of (**a**) 40 mW; (**b**) 60 mW; (**c**) 100 mW. (**d**) Quantification and (**e**) box plot of the radius of the different AuNSs printed. Box plots were made in all cases from at least 180 independent measurements (*N* > 180), as shown in (**d**).

**Figure 3 ijms-22-07465-f003:**
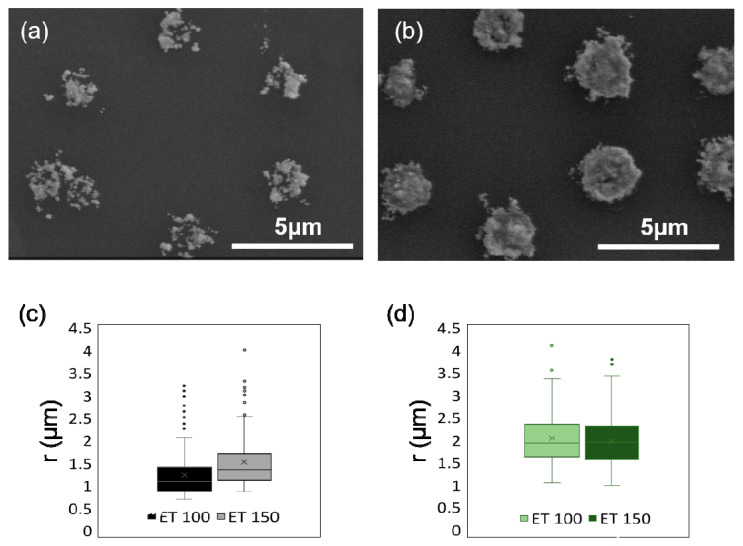
SEM micrographs showing AuNSs fabricated via TPA–DLW using an ET = 150 ms and (**a**) LP = 40 mW; (**b**) LP = 100 mW. Box plots comparing the radius of the different AuNSs printed with different ET at (**c**) LP = 40 mW and (**d**) LP = 100 mW. Box plots were made in all cases from at least 180 independent measurements (*N* > 180).

**Figure 4 ijms-22-07465-f004:**
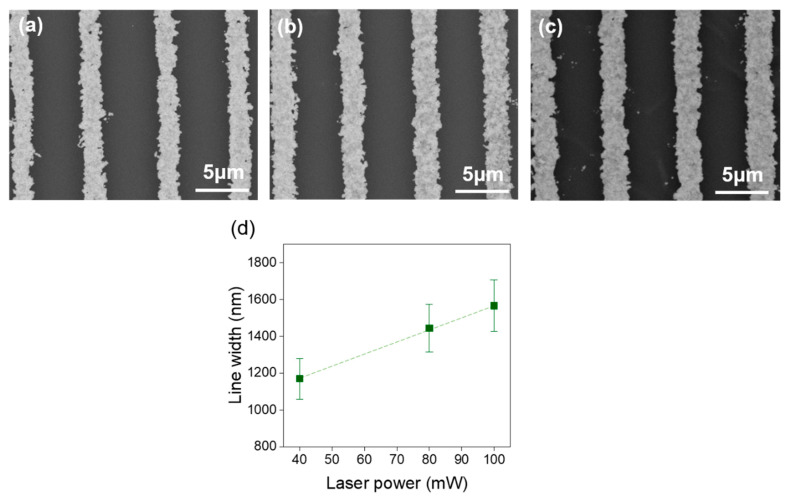
SEM micrographs showing AuNSs fabricated via DLW using an SS = 200 µm/s and an LP of (**a**) 40 mW; (**b**) 60 mW; (**c**) 100 mW. (**d**) Average line width of the different AuNSs printed as a function of the LP.

**Figure 5 ijms-22-07465-f005:**
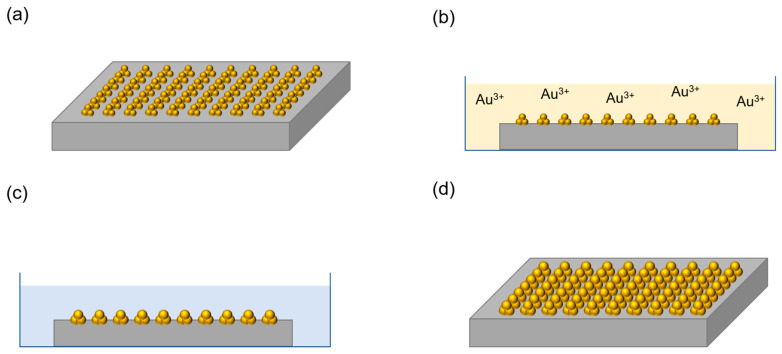
Scheme depicting the nucleation and the growth of the AuNSs created. (**a**) the first step is the creation via TPA–DLW of the AuNSs on the silicon substrate. After TPA–DLW, the AuNSs keep growing until a washing step with deionized water is carried out to dilute the solution and subsequently to remove the hydrogel; (**b**) then, in order to tune the size of the AuNSs, a second immersion in a 10^−2^ M HAuCl_4_ solution is completed. This allows a controlled growth over time of the AuNSs through the migration of the Au^3+^ ions towards the nanostructures and their reduction into Au^0^ on the surface of the AuNSs by autocatalytic reduction; (**c**) a subsequent bath in deionized water stops the growth; (**d**) the AuNPs have grown in a controlled way as a function of the immersion time only where the AuNSs are lodged.

**Figure 6 ijms-22-07465-f006:**
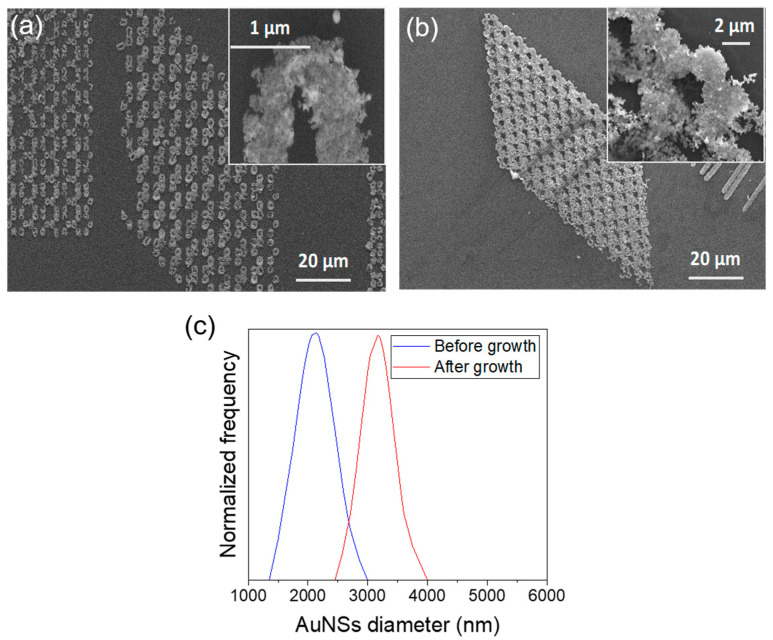
SEM images of the nanopatterns (**a**) before and (**b**) after AuNSs’ growth by immersion in a 10^−2^ M HAuCl_4_ bath for 30 min. (**c**) Distribution of the AuNSs’ size before growth (blue, *N* = 29) and after growth (red, *N* = 26).

**Figure 7 ijms-22-07465-f007:**
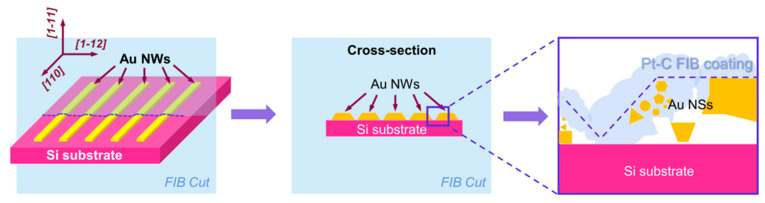
Scheme of the sample preparation for TEM analysis via FIB cut. The sample was studied along the Si (substrate) (110) zone axis. A layer of Pt is deposited on the top of the sample with an ion beam to protect the structures from the destructive cutting procedure.

**Figure 8 ijms-22-07465-f008:**
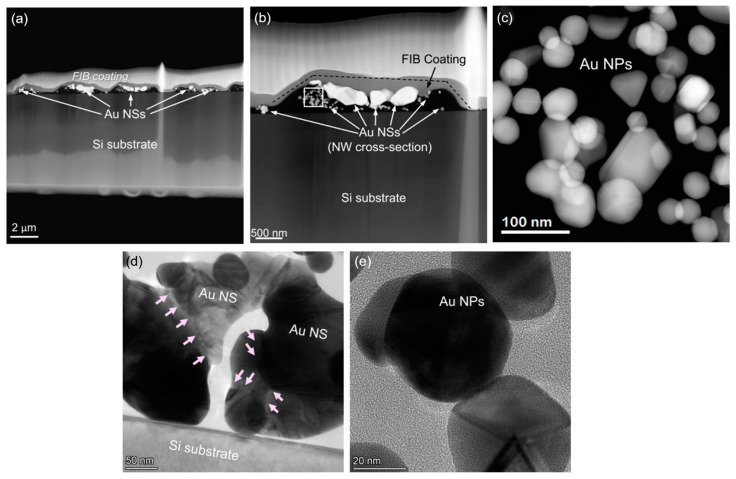
(**a**–**c**) HAADF and (**d**,**e**) HRTEM images of the cross-section of different AuNSs made with an LP = 100 mW and ET = 150 ms after TPA–DLW. These images illustrate the autocatalytic growth of the AuNSs after TPA–DLW, before the HAuCl_4_ bath (see Figure 1d or Figure 5a). The duration of this growth is estimated as the lag between the TPA–DLW process and the washing step with deionized water (~10 s); AuNPs appear brighter than the background in the HAADF images (**a**–**c**) since the regions with heavier elements provide a more intense signal in this mode; (**b**,**d**) show that the AuNSs are physically deposited onto the Si substrate. Arrows in (**d**) indicate merging regions since TPA–DLW is able to create larger structures by the coalescence of smaller individual AuNPs. Different shapes of the individual AuNPs (decahedral, pentagonal bipyramid) are shown in (**c**,**e**).

**Figure 9 ijms-22-07465-f009:**
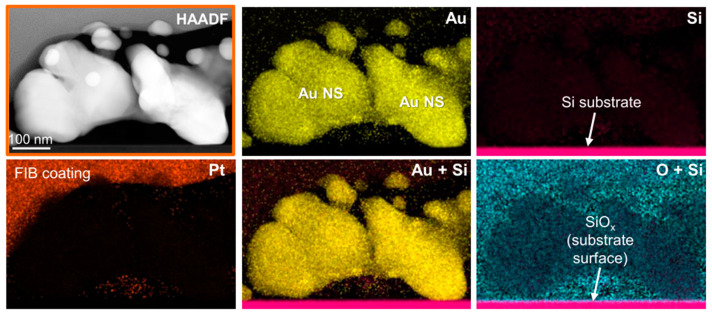
HAADF image and EDX maps of AuNSs made with an LP = 100 mW and ET = 150 ms. Au, Si, Pt and O signals are displayed in yellow, magenta, orange, and cyan, respectively. AuNSs consist of individual AuNPs, which self-assemble into larger, micron-sized clusters. These clusters contain only Au, without any impurities (e.g., oxidized species or precursor residues). EDX also shows a thin SiOx native layer at the substrate surface (~2 nm thick).

**Figure 10 ijms-22-07465-f010:**
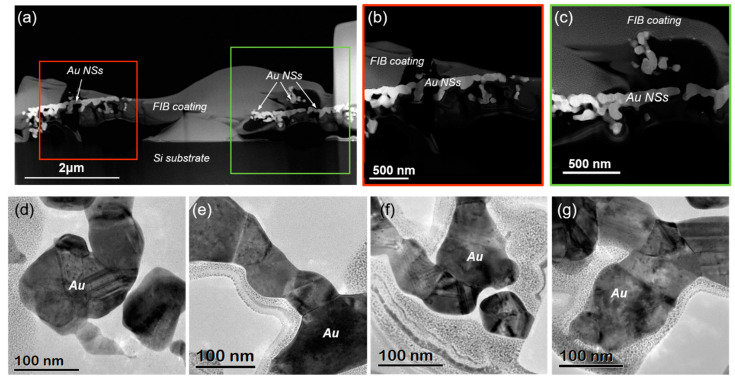
(**a**–**c**) STEM after FIB cut and (**d**–**g**) TEM images of the cross-section of different AuNSs made with an LP = 100 mW and ET = 150 ms and the second immersion in a 10^−2^ M HAuCl_4_ solution for 30 min. After the bath (see Figure 5d), individual AuNPs are not appreciated. High magnification TEM images of the AuNPs in (**d**–**g**) show that they have been merged by their facets and exhibit a higher polycrystalline morphology.

**Figure 11 ijms-22-07465-f011:**
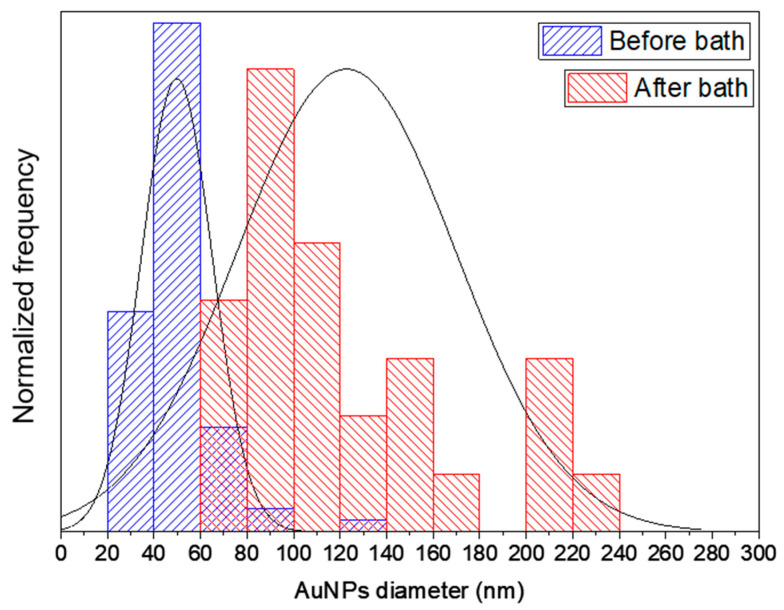
AuNPs size distribution after TPA–DLW and immersion in deionized water (blue, *N* = 75) and after immersion in a 10^−2^ M HAuCl_4_ bath for 30 min (red, *N* = 27). Gaussian distribution, before and after the bath, evidence the growth of the AuNSs. Quantitative analysis of the AuNPs’ sizes shows that they range from 50 ± 11 nm to 123 ± 35 nm.

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
