# Peer review of "Additive Manufacturing of Gold Nanostructures Using Nonlinear Photoreduction under Controlled Ionic Diffusion"

_ijms, 2021, doi:10.3390/ijms22147465_

Round 1

Reviewer 1 Report

Di Cianni and co-workers submitted the manuscript entitled “Additive Manufacturing of Metallic Nanostructures using Non-linear Photoreduction under Controlled Ionic Diffusion”, to be published in “International Journal of Molecular Sciences (I.F = 4.556)”. In this paper author utilized the Direct Laser Writing (DLW) Two Photons Absorption (TPA) process for micro and nanofabrication of gold nanostructures (AuNSs) in a natural polymeric matrix (isinglass). Overall, this is a nice work, but need to undergo major revision before its publication.

  1. In introduction author must justify why they intend to use HAuCl4 as precursor and natural polymeric matrix (isinglass), the importance of them also needs to be discussed for the new readers. Author also needs to justify this proposed tactic can be applied to other metallic nanostructures other than AuNSs as prescribed in the title.
  2. SEM images 2(a), 2(b) and 3(a) does not have any scale bars, without the scale bars, it is not possible to confirm the stated information is correct or not?
  3. Explanation of Figure 5 must be upgraded, which justifies the growth of the AuNSs. Currently, this is an incomplete section.
  4. Explanation for Figures 8-11 must be boosted in conjunction with Figure 5, this is also looking like an incomplete section.
  5. Author must provide a paragraph that compare the merits the proposed tactics rather than the available other processes for AuNSs.
  6. Number of measurements (n=?) must be provided for all box plots.
  7. Details of instruments used must be provided in Materials and methods.
  8. Reference section requires updation for DLW and TPA.     

Author Response

Reviewer #1

Di Cianni and co-workers submitted the manuscript entitled “Additive Manufacturing of Metallic Nanostructures using Non-linear Photoreduction under Controlled Ionic Diffusion”, to be published in “International Journal of Molecular Sciences (I.F = 4.556)”. In this paper author utilized the Direct Laser Writing (DLW) Two Photons Absorption (TPA) process for micro and nanofabrication of gold nanostructures (AuNSs) in a natural polymeric matrix (isinglass). Overall, this is a nice work, but need to undergo major revision before its publication.

  1. In introduction author must justify why they intend to use HAuCl4 as precursor and natural polymeric matrix (isinglass), the importance of them also needs to be discussed for the new readers. Author also needs to justify this proposed tactic can be applied to other metallic nanostructures other than AuNSs as prescribed in the title.

We have enlarged the introduction in this regard. We have also explained more in detail the advantages of isinglass at the beginning of the results section. We have modified the title and changed “metallic nanostructures” by “gold nanostructures”. We believe that it is more accurate now.

  1. SEM images 2(a), 2(b) and 3(a) does not have any scale bars, without the scale bars, it is not possible to confirm the stated information is correct or not?

We have included the scale bars in all the images as suggested by the reviewer.

  1. Explanation of Figure 5 must be upgraded, which justifies the growth of the AuNSs. Currently, this is an incomplete section.

We have modified the legend of Figure 5. We hope it is clearer now.

  1. Explanation for Figures 8-11 must be boosted in conjunction with Figure 5, this is also looking like an incomplete section.

We have modified the legends of Figures 8-11. We hope it is clearer now.

  1. Author must provide a paragraph that compare the merits the proposed tactics rather than the available other processes for AuNSs.

We have added a paragraph about this in the introduction.

  1. Number of measurements (n=?) must be provided for all box plots.

We have included the number of measurements in the legend of the figures in all cases.

  1. Details of instruments used must be provided in Materials and methods.

We have added the details about the TPA-DLW instruments in the Materials and Methods section. This section can be found before the Conclusions section, as recommended by the guidelines of the IJMS template.

  1. Reference section requires updation for DLW and TPA.

We have added more recent literature, as suggested by the reviewer. Please, see refs. 9, 10, 11, 28, 38, 39, 40 and 41.

Reviewer 2 Report

Nanoparticles of various sizes are used in the development of test systems and are usually particles in the nanoscale range. The authors showed the possibility of obtaining even larger ones and characterized the obtained preparations.

  1. Figure 3. The obtained nanoparticles are large in size. What explains such a large size? after all, the method of chemical synthesis allows one to obtain small particles from 1 to 50-150 nm, as well as the ablation method.
  2. Page 8 of 15, Line 213. HAADF needs the full name before the abbreviation.
  3. Please check all abbreviations along the manuscript. At the place of the first mention, you must enter the abbreviation and then only use it.

Author Response

Reviewer #2

Nanoparticles of various sizes are used in the development of test systems and are usually particles in the nanoscale range. The authors showed the possibility of obtaining even larger ones and characterized the obtained preparations.

  1. Figure 3. The obtained nanoparticles are large in size. What explains such a large size? after all, the method of chemical synthesis allows one to obtain small particles from 1 to 50-150 nm, as well as the ablation method.

Along the whole manuscript, we stablish a difference between gold nanostructures (AuNSs) and gold nanoparticles (AuNPs). What it is shown in Figure 3 are the AuNSs synthesized by TPA-DLW. These AuNSs are formed by smaller, individual AuNPs. We study this in more detail with TEM (see for instance, Figure 8), where we show that these AuNSs are composed of individual AuNPs with average sizes of 50 nm. TPA-DLW is a nanofabrication technique which allows to fabricate 2D and 3D structures immobilized onto a solid substrate. This approach is different to chemical synthesis which allows to obtain AuNPs homogeneously dispersed in solution.

  1. Page 8 of 15, Line 213. HAADF needs the full name before the abbreviation.

We have defined and changed this accordingly.

  1. Please check all abbreviations along the manuscript. At the place of the first mention, you must enter the abbreviation and then only use it.

We thank the reviewer for this observation. We have revised all the abbreviations.

Round 2

Reviewer 1 Report

Author must improve the resolution of Figures

Reviewer 2 Report

The authors adequately responded on all questions and thus this manuscript can be accepted in its current form